# Didecyldimethylammonium Chloride- and Polyhexamethylene Guanidine-Resistant Bacteria Isolated from Fecal Sludge and Their Potential Use in Biological Products for the Detoxification of Biocide-Contaminated Wastewater Prior to Conventional Biological Treatment

**DOI:** 10.3390/biology11091332

**Published:** 2022-09-09

**Authors:** Nataliya Loiko, Oleg Kanunnikov, Dmitriy Serdyukov, Vladimir Axelrod, Eduard Tereshkin, Anastasia Vishnyakova, Yuriy Litti

**Affiliations:** 1Winogradsky Institute of Microbiology, «Fundamentals of Biotechnology» Federal Research Center, Russian Academy of Sciences, 117312 Moscow, Russia; 2Rail Chemical LLC, 105005 Moscow, Russia; 3Semenov Federal Research Center for Chemical Physics, Russian Academy of Sciences, 119991 Moscow, Russia

**Keywords:** biocide, fecal sludge, environmentally safe toilet complexes, *Alcaligenes faecalis*, didecyldimethylammonium chloride, polyhexamethylene guanidine, microbial resistance, biological products, detoxification

## Abstract

**Simple Summary:**

Every year, more than a million tons of fecal sludge (FS) containing biocides based on quaternary ammonium compounds and guanidine derivatives, which are widely used for FS deodorization and control of microbial activity, are generated in the environmentally safe toilet complexes of Russian Railways trains. Higher disposal costs for such biocide-contaminated FS due to activated sludge toxicity increases pressure on sanitary equipment servicing companies («Ecotol Service» LLC) to more efficiently discharge FS to wastewater treatment plants. In this work, we have developed a new environmentally friendly approach to reducing the toxicity of FS, based on the use of biological products from biocide-resistant bacterial strains isolated from FS. Our approach has proven to be effective in a series of FS biodegradation experiments, biological oxygen demand tests, and a newly developed disk-diffusion bioassay.

**Abstract:**

Toxic shock caused by the discharge of biocide-contaminated fecal sludge (FS) from chemical toilets to conventional wastewater treatment plants (WWTP) can be a major problem in activated sludge operation. It is necessary to develop new environmental approaches to mitigate the toxicity of biocides in order to avoid degrading the performance of WWTP. “Latrina”, a chemical toilet additive containing didecyldimethylammonium chloride and polyhexamethylene guanidine, is widely used in environmentally safe toilet complexes (ESTC) on Russian railway trains to deodorize FS and control microbial activity. In this work, seven biocide-resistant bacterial strains were isolated and identified from the FS of ESTC. The values of the minimum inhibitory and bactericidal concentrations of biocides for the isolated strains were 4.5–10 times higher than for the collection microorganisms. The bacterium *Alcaligenes faecalis* DOS7 was found to be particularly resistant to “Latrina”, the minimum inhibitory concentration of which was almost 30 times higher than recommended for ESTC. Biological products based on isolated bacterial strains proved to be effective for FS biodegradation under both aerobic and anaerobic conditions. The results of the biochemical oxygen demand test and the newly developed disk-diffusion bioassay confirmed that isolated strains contribute to reducing toxicity of biocidal agents in FS. Hyper-resistance, non-pathogenicity, and potential plant growth-promoting ability make *A. faecalis* DOS7 promising for use in various biological products for wastewater treatment and bioremediation of soils contaminated with biocides, as well as in agriculture to increase plant productivity.

## 1. Introduction

Due to the ongoing large-scale urban development, domestic wastewater is one of the largest sources of water pollution, so the problem of wastewater treatment is becoming more acute and relevant throughout the world [1,2]. The number and variety of pollutants, toxic substances with chemical resistance, biotoxicity and the ability to bioaccumulate entering the aquatic environment is constantly growing. This is alarming as widespread pollution of water and the environment threatens the sustainability of ecosystems and human health [3,4]. Wastewater consists of so-called gray water, which is formed during washing, bathing, etc., and black water coming from toilets [2,5]. In this regard, wastewater contains a large amount of solids, organic and inorganic compounds in dissolved or suspended forms, heavy metals, microplastics, biocides, pharmaceutically active compounds, antibiotics and other biologically active substances that pollute the environment [6,7,8,9,10,11].

The presence of biocidal compounds such as quaternary ammonium compounds (QACs) and biguanide derivatives in wastewater is of increasing concern. QACs, such as didecyldimethylammonium chloride (DDAC) and alkyldimethylbenzylammonium chloride (ADBAC) are a very popular group of cationic surfactants. They have a hydrophilic polar head and hydrophobic non-polar alkyl ends [12]. Due to this structure, QACs are able to effectively concentrate on the surface or at the interface, significantly reducing the surface tension [13]. The same feature of the QAC structure is also responsible for their antimicrobial effects [14,15,16]. These substances are biocidal against a number of microorganisms, including fungi, Gram-negative and Gram-positive bacteria, and lipophilic viruses [17]. QAC easily integrate into cell membranes, and also inhibit the activity of proteins (H^+^-ATPase) and disrupt the transport of substances to the cell. Moreover, in the presence of QAC, changes in the lipid composition (qualitative and quantitative) of the plasma membrane are observed. Their biocidal activity against microorganisms such as bacteria, fungi, and viruses with a lipid membrane depends on the chemical structure of the compound [18,19]. The highest antimicrobial activity is exhibited by QAC with two or more charges (bis-QAC, multi-QAC, poly-QAC), which have an alkyl tail length in the range of C10–C14 [20]. For the first time, QAC appeared on the disinfectant market in the 1930s after the discovery of the antimicrobial properties of benzalkonium chloride [21]. Currently, it is one of the most popular biocidal agents, the world production of which exceeds 500,000 tons per year [22].

Along with QAC, mono- and biguanide derivatives have become widespread in medicine, industry and household use [23,24,25,26,27,28,29]. The most famous of them include chlorhexidine digluconate (CHG), polyhexamethylene guanidine (PHMG) and polyhexamethylene biguanide (PHMB). As well as QAC, mono- and biguanides are amphipathic compounds, cationic in nature with a similar mechanism of action on microorganisms. Attaching to lipids inside the membrane of the target cell, they nonspecifically destroy its components, causing leakage of the cytoplasmic contents of the bacterium [30]. It was also shown in a number of works that PHMB can penetrate into bacterial cells and cause condensation of chromosomes [31]. At the same time, PHMB can also penetrate into mammalian cells, but there it is captured by endosomes and expelled from the nuclei [30].

QACs are now widely used in various consumer and commercial products, medicine (disinfectants, medicines), chemical and cosmetic industries (preservatives, rinses, shampoos and hair conditioners), agriculture and forestry (insecticides, fungicides) [32,33,34,35,36,37]. The broad antimicrobial specificity of guanides and a low level of toxicity to humans made it possible to use these substances in the food industry, healthcare (for disinfection of wounds, dressings and dishes), and in the household for disinfection of water and surfaces [38,39,40]. Biguanides such as metformin hydrochloride are used to treat type 2 diabetes. Their action as an antidiabetic drug is to reduce the production of glucose in the liver by inhibiting gluconeogenesis, as well as by increasing insulin-stimulated glucose uptake in skeletal muscle [41].

One of the new uses for these biocidal agents is as a component of a chemical toilet additive designed to deodorize fecal waste and control microbial activity. The combined use of QAC and PHMG as biocides to prevent the biodegradation of human waste products by saprotrophic microorganisms and eliminate an unpleasant odor in fecal sludge (FS), for example, in environmentally safe toilet complexes (ESTC) of passenger railcars, has the task of preserving human waste products until their subsequent processing or disposal [42,43]. However, when disposed of in municipal wastewater treatment plants (WWTPs), such waste poses a potential threat, as it negatively affects activated sludge, inhibiting the processes of nitrification, denitrification, and biochemical oxygen demand (BOD) [44,45]. Therefore, before their disposal, preliminary neutralization of the toxic effect of biocides should be carried out. There is still very little development in this area. Previously, Litti et al. [42] proved the effectiveness of the deactivating agent and chemical neutralization by strongly acidifying or alkalizing the medium to deactivate QAC and PHMG. However, the use of additional chemicals increases processing costs.

Currently, the development of microbiological methods for the decontamination of wastewater is of paramount interest. For example, biological products, including living and nonliving microorganisms (bacteria, algae, fungi, and yeast), are used to remove and recover toxic or precious metals from industrial wastewater [46]. These methods are effective due to their availability and low cost, and are more environmentally friendly than the use of chemical methods. The use of biological products can not only contribute to the removal of various contaminants, but also increase the efficiency of WWTPs due to enhanced biodegradation.

Biological products are a single culture or a consortium of microorganisms, often represented by various aerobic and facultative anaerobic bacteria belonging to the genera *Bacillus*, *Pseudomonas*, *Ruminococcus*, *Bacteroides*, *Lactobacillus*, etc. The activity of these microorganisms contributes to the solubilization (liquefaction) of waste [47]. Biological products often contain microbial enzymes such as protease, amylase, lipase, and cellulase, as well as various mineral additives that stimulate microbial growth [48,49]. Microbial cultures that are part of biological products must meet basic requirements, including: high growth rate, sufficient level of enzyme secretion, the ability to adapt and maintain viability and functionality in a wide range of pH, temperatures, as well as in the presence of detergents and household chemicals.

This study was aimed at finding new isolates with high resistance to QAC and PHMG, able to reduce the toxic effects of these biocidal agents for the effective biological treatment of fecal waste.

## 2. Materials and Methods

### 2.1. Fecal Sludge and Biocides

Liquid fecal sludge (FS) was sampled at the beginning of June 2019 from the environmentally safe toilet complexes (ESTC) of railway trains of the North-Western branch of the Joint-stock company «Federal Passenger Company» (JSC «FPK», Moscow, Russia). FS contained 0.07% of the biocide “Latrina” produced by Limited Liability Company «Rail Chemical» (Moscow, Russia). “Latrina” is widely used in the ESTC of JSC «FPK» and by Russian Railways as toilet chemical additive. The composition of “Latrina” included: DDAC (0.24%) and PHMG (6.5%) (total 6.74%), surfactants, perfume and water. For biological oxygen demand tests, FS without “Latrina” treatment was additionally obtained from the ESTC of JSC «FPK».

Individual GACs and guanidine and biguanide derivatives were also used in the work:-20% aqueous solution of PHMG hydrochloride (biocide “Biopag D”, International Institute of Ecological and Technological Problems, LLC, Moscow, Russia);-50% aqueous solution of ADBAC (biocide “Catamine AB”, Npao NPF Bursintez-M, Moscow, Russia);-50% aqueous isopropanol solution of DDAC (biocide “SEPTAPAV HSV.50”, Scientific Manufacturing Enterprise «NIIPAV», Volgodonsk, Russia)-20% aqueous solution of CHG (biocide “Dezin”, “Dezindustriya” LLC, Moscow, Russia).

### 2.2. Isolation of GAC- and PHMG-Resistant Microorganisms

FS sample was plated on an agar LA nutrient medium of the following composition (g/L): yeast autolysate—10.0; peptone—5.0; NaCl—5; agar 3; pH 7.0. Petri dishes were incubated at a temperature of 28 °C for a day and the fastest growing colonies were isolated into pure cultures by the method of multiple passages. All microbiological experiments were performed under sterile conditions in a laminar flow box.

### 2.3. Identification of Isolated Microorganisms

DNA was isolated from microbial cultures according to the previously described method [50]. Libraries of 16S rRNA fragments were prepared by amplification according to [51]. Primers UNIV-515F–UNIV-806R were used as the sequences flanking V3 [52]. For PCR, 5 × Taq Red buffer and HS Taq polymerase (Evrogen, Moscow, Russia) were used according to the manufacturer’s recommendations. The final concentration of primers in all cases was 1 μM. The final reaction volume was 30 μL. Each DNA sample was amplified in triplicate. Then the replicates were pooled and visualized on a 2% agarose gel at a wavelength of 470 nm. The desired DNA band was excised and purified using the Standard Cleanup Gel Extraction Kit (Evrogen, Moscow, Russia). To measure the DNA concentration, a Qubit^®^ 2.0 fluorometer with an HS Assay Kit (Life Technologies, Carlsbad, CA, USA) was used. Before sequencing, the samples were mixed equimolarly and the resulting DNA solution was diluted to 4 nM. Further denaturation of the library pool and preparation for sequencing were performed according to the standard Illumina Sample Preparation Guide protocol on the MiSeq platform using the MiSeq Reagent Kit v2 (500 cycles) (Illumina, San Diego, CA, USA). The primary data processing, the formation of the OTE table, and the analysis of the taxonomic composition and alpha metrics of diversity were performed using QIIME (version 1.9.1) [53] and the SILVA online data analysis service [54].

### 2.4. Cultivation

The bacteria (isolated and collection strains) were grown in lysogeny-broth medium (LB) (Broth, Miller, VWR Life Science, United States). They were cultivated in 250 mL flasks with 50 mL of nutrient medium with mixing on an orbital shaker (120 rpm) for 24 h (to the stationary phase) at a temperature of 28 °C. Inoculum (culture at the beginning of the stationary growth phase) was introduced in an amount of 0.25 mL per 50 mL of medium (0.5% vol.).

### 2.5. Determination of Minimum Inhibitory (MIC) and Bactericidal (MBC) Concentrations

Biocide aliquots were added to 25 mL glass test tubes with cotton stoppers containing 5 mL of LB culture medium, varying their final concentrations in the medium. The tubes were then inoculated with microorganisms at stationary growth phase and incubated in a thermostatically controlled shaker (28 °C, 120 rpm). After 2 days of incubation, the growth of microorganisms in the presence of various biocide concentrations was assessed visually by the appearance of turbidity. The lowest concentration of the biocide, at which no growth of the test cultures was observed, was taken as the MIC. Moreover, after 2 days of incubation, aliquots were plated on an agar LA nutrient medium from test tubes, in which no visual microbial growth was observed. The MBC was taken to be the lowest concentration of the biocide in the test tube, at which no growth of microorganisms was observed on the agar medium.

### 2.6. Evaluation of the Effect of the Biocide “Latrina” on Bacterial Growth

To study the effect of “Latrina” on the growth of 7 isolated strains, they were grown on an LB medium of the following composition (g/L): yeast autolysate—10.0; peptone—5.0; NaCl—5; pH of the medium 7.0. The cultivation was carried out in 25 mL test tubes with 5 mL of nutrient medium with stirring on an orbital shaker (120 rpm) at a temperature of 28 °C for 2 days. Inoculum—a culture of the beginning of the stationary phase (overnight culture), was introduced in an amount of 50 μL per 5 mL of medium (1% vol.). Before the introduction of the inoculum, “Latrina” was added into the test tubes to a final concentration of 50, 100 and 200 mg/L (in the case of the 7th strain (Table 1), the range of biocide concentrations was expanded). No biocide was added to control tubes. At the end of the cultivation, viable cells were counted in each experimental tube, plating aliquots after the appropriate dilution on agar LA nutrient medium. Petri dishes were incubated at a temperature of 29 °C for 3 days, after which the number of colony forming units (CFU) in the corresponding dilutions was counted and the cell number (CFU/mL) in the experimental tube was calculated.

### 2.7. Preparation of Biological Products Based on Isolated Bacterial Strains

Bacteria were grown in LB medium, as described above. Bacterial culture was then mixed with sterile ground rice husk (less than 2 mm fraction) in a ratio of 1/0.5 by weight and dried on a tray drier at a temperature of 36 °C. The dry biological products contained (1.0 ± 0.5)·10^10^ cells/g.

### 2.8. Evaluation of the Effect of Biological Products on the Biodegradation of FS

First, 80 mL of FS was added to 265 mL glass bottles. The pH of the FS was adjusted to 7.0 by adding 5N HCL solution. Then, 0.5 g of biological product was added to FS. For anaerobic incubation experiments, the bottles were thoroughly purged with nitrogen. The bottles were closed with rubber stoppers and crimp cap, and incubated in dark at a temperature of 23 °C. After 7 days of incubation, the number of viable cells was determined as described above.

Under anaerobic conditions, the rate of oxygen consumption was taken as the criterion for the rate of FS biodegradation, and under anaerobic conditions, the rate of carbon dioxide accumulation. After 5 days of incubation, FS samples (35 mL) were taken to determine the five-day biochemical oxygen demand (BOD_5_). This did not disturb the experimental process, since additional replicates were made for each determination of BOD_5_.

### 2.9. Analytical Methods

The oxygen and carbon dioxide concentrations in the headspace of the bottles were determined by gas chromatograph Crystal 5000.2 (Chromatec, Yoshkar-Ola, Russia) as described earlier [55]. BOD_5_ was determined by the OxiTop respirometric BOD measuring system (WTW, Oberbayern, Germany) according to manufacturer’s recommendations.

A disk-diffusion bioassay has been developed to assess the residual toxicity of the biocide in FS. As microbiological test models, typical representatives of soil and FS were used: Gram-positive bacteria Micrococcus luteus NCIMB 13267 and yeast Yarrowia lipolytica 367-2. Microorganisms were grown as described above until the stationary growth phase. Then, an aliquot (0.1 mL) of a microbial suspension containing 10^8^ cells was transferred to a Petri dish with LB agar medium and evenly distributed over the entire surface of the dish with a spatula. Then, paper disks (Whatman 2017-006 AA Disks 1000 × 6 mm Antibiotic Assay Disks, Whatman, Maidstone, UK-Germany) were placed on the agar, onto which 20 µL of FS was applied. After incubation of the plates in a thermostat at 28 °C for 48 h, the growth inhibition zone around the disks was determined. The bactericidal effect of the FS was evaluated by the zone diameter (in mm).

### 2.10. Statistical Methods

All experiments were performed in triplicate. Statistical analysis was carried out using standard mathematical methods (Student’s *t*-test and calculation of the standard deviation) using the Microsoft Excel program. The data group was considered homogeneous if the mean square deviation σ did not exceed 10 per cent. The differences between the data groups were considered valid under the probability criterion *p* < 0.05.

## 3. Results

### 3.1. Isolation and Identification of Biocide-Resistant Microorganisms from FS

The addition of biocide “Latrina” (700 ppm) containing DDAC and PHMG (6.74% in total) to FS inhibits both aerobic and anaerobic microorganisms [15]. However, FS may also contain microorganisms that are resistant to these biocides. The addition of DDAC- and PHMG-resistant microorganisms to the FS before it is discharged to WWTP may improve the biodegradation of the FS and reduce toxicity.

To isolate DDAC- and PHMG-resistant microorganisms, FS was plated on agar LB medium. After a day of incubation, the first colonies appeared. The number of viable cells in FS, calculated after inoculation on solid medium, was (3.6 ± 0.3)·10^6^ CFU/mL. For further work, seven different colonies were selected, which were the most common and the first to appear on the agar medium (Figure 1). Of these, after standard purification procedures, seven bacterial strains were isolated into a pure culture and identified by standard molecular genetic methods (by analysis of 16S rRNA). The strains were deposited at Genbank under the accession numbers shown in Table 1.

The isolated bacterial cultures grew well on standard microbiological media under aerobic cultivation conditions with high cell numbers. The storage of these cultures was possible in various ways, both in a lyophilic state and at low temperatures with the addition of glycerol as a cryoprotectant. The isolated bacterial strains were sent for testing at the Research Center for Toxicology and Hygienic Regulation of Biopreparations (http://toxicbio.ru/ accessed on 5 August 2021). Based on the test results, strains DOS 1–7 were not pathogenic for warm-blooded animals in terms of virulence, dissemination, toxicity and toxegenicity.

### 3.2. Resistance of the Isolated Strains to the Biocide “Latrina” Containing DDAC and PHMG

The resistance of the isolated bacterial strains to the biocide “Latrina” containing QAC and PHMG turned out to be quite high. In three cultures out of seven, the values of the minimum inhibitory concentration (MIC) were higher than the concentration of DDAC and PHMG used to control microbial activity in FS (47.25 ppm in total) (Table 2). In addition, the values of MIC and minimum bactericidal concentration (MBC) in relation to the isolated strains of *Staphylococcus, Enterococcus, Micrococcus and Bacillus* were 4.5–10 times higher than in relation to their collection counterparts (Table 2). The bacterium *Alcaligenes faecalis* DOS7 turned out to be especially resistant to the biocide, the MIC for which was almost 30 times higher than the concentration of DDAC and PHMG normally added to FS with “Latrina”. Of the collection cultures, only cells of the Gram-negative bacterium *Pseudomonas aeruginosa* 4.8.1 demonstrated high resistance to DDAC and PHMG. Different MIC and MBC of PHMG for some bacteria have been reported: e.g., 2–8 ppm for *Staphylococcus aureus*, 4–8 ppm for *Enterobacter* spp.; 4–32 ppm for *Enterococcus faecium* and 16–32 ppm for *Pseudomonas aeruginosa* [56].

In the presence of “Latrina” in a liquid nutrient medium at concentrations below the MIC (with a total content of DDAC and PHMG of 3.5, 7.0 and 14.0 ppm), a steady growth of the isolated strains was observed, however, the growth was less pronounced than in the control treatments without the addition of “Latrina”. Table 3 shows the cell number in populations of seven isolated strains that grew in the presence of different concentrations of biocide and without it. In the first six bacterial cultures at a total concentration of DDAC and PHMG of 3.5 ppm, the number of cells after two days of cultivation was 5–10 times less than in the control population (Table 2). An increase in the biocide content in the nutrient medium led to an even greater decrease in cell number.

Cultivation of the seventh strain, *A. faecalis* DOS7, which was especially resistant to the action of DDAC and PHMG, showed that in a wide range of biocide concentrations (14–240 ppm), the number of cells after two days of cultivation still remained at a high level, corresponding to 20–30% of the control population (Figure 2).

In addition to the biocide “Latrina”, *A. faecalis* DOS7 has also shown high resistance to some individual GACs and guanidine and biguanide derivatives (Table 4). Surprisingly, MIC and MBC of “Latrina” turned out to be higher than of its individual biocidal components (DDAC and PHMG). This may be due to the presence of other surfactants in the composition of “Latrina” (in addition to PHMG and especially DDAC, which also have the ability to lower the surface tension), since it has been shown that in the presence of anionic surfactants, QACs become less effective [57]. Thus, the demonstrated resistance of the isolated bacterial cultures to the action of DDAC and PHMG provides a basis for obtaining promising biological products from them.

### 3.3. Biological Products Based on Isolated Bacterial Strains and Their Effect on the Bio-Degradation of FS

Biological products with a high cell number (1.0 ± 0.5)·10^10^ CFU/g) were prepared on the basis of seven isolated strains. Biological products were named after the name of the corresponding strain: DOS1, DOS2, DOS3, DOS4, DOS5, DOS6, DOS7. They were tested for the efficiency of biodegradation of organic matter in FS. Biological products were introduced into FS (0.5 g per 80 mL), the mixture was incubated for 7 days at a temperature of 25 °C. The process of decomposition of organic matter of FS was evaluated both under aerobic and anaerobic conditions. Under aerobic conditions, the rate of oxygen consumption (O_2_) was taken as the criterion for the efficiency of FS biodegradation, and under anaerobic conditions, the rate of accumulation of carbon dioxide (CO_2_). The experimental treatments were compared with the control treatment without the addition of biological products to FS (Table 5).

The addition of biological products to the FS accelerated oxygen consumption under aerobic conditions by more than 2 times, and also contributed to an increase in the rate of carbon dioxide accumulation by 17–22 times under anaerobic conditions. The greatest effect was achieved under aerobic conditions when using DOS7 and under anaerobic conditions when using DOS4.

It should be noted that under anaerobic conditions, the formation of CO_2_ stopped on day 3–4 of the experiment, which is possibly associated with the accumulation of metabolites of anaerobic decomposition, which could not be further converted into methane due to the absence of the corresponding group of microorganisms in FS. Under aerobic conditions, complete oxygen consumption already occurred on days 1–2, therefore, during the experiment, it was additionally added three times by purging with air.

The number of bacteria added with the biological product in each case averaged 6.25·10^7^ cells/mL. As a result, after seven days of the aerobic treatment, the cell number in treatments with the addition of DOS 1, 4, 5, 6 was at the same level. In treatments with the addition of DOS 2 and 3, it decreased by about 4 times. In the case of using DOS7, the number of viable bacteria at the end of the experiment became an order of magnitude higher, which indicated the activation of microbiological processes in FS.

During the anaerobic biodegradation of FS, apparently, some of the bacteria introduced with the bacterial products passed into a dormant (non-cultivation) state, therefore, they were not detected at the end of the experiment. However, in the case of DOS7, we again found an increase in the number of cells by 36%.

Thus, it was shown that the biocide “Latrina” has a strong antimicrobial effect and is able to inhibit the activity of the aerobic and anaerobic microflora of FS. However, the introduction of biological products based on isolated bacterial strains enhanced the processes of aerobic and anaerobic biodegradation of FS containing DDAC and PHMG.

### 3.4. Effect of Bacterial Products on Detoxification of the Biocide “Latrina” in FS

Intensification of FS biodegradation processes should contribute to the neutralization of the biocidal action of DDAC and PHMG, the active components of the biocide “Latrina” added to the PS. The assessment of the decrease in the toxicity of “Latrina” in the FS was carried out in two ways: (1) using a test to determine the respiratory activity of microorganisms during the incubation of FS samples for 5 days under conditions favorable to microbial activity; and (2) using the method of imposing disks on test cultures.

In the first case, the BOD_5_ value was obtained using the method for determining the quantity of dissolved oxygen consumed in 5 days by biological processes breaking down organic matter of FS samples at a temperature of 20 °C without access to light. In non-treated FS (Control 1: FS without added Latrina), the BOD_5_ value was the highest because aerobic microorganisms were not inhibited. The BOD_5_ test using FS treated with the biocide “Latrina” (700 ppm) and incubated for 7 days under aerobic or anaerobic conditions (Control 2: FS with added Latrina) showed a decrease in the respiratory activity of microflora by 2.8 and 1.3 times, respectively. However, when new biological products were added, these values were close to the Control 1 level, which indicated the neutralization of the toxicity of the biocide in relation to the biodegradation of FS. The greatest effect was observed for the samples after incubation under aerobic conditions with the use of bacterial products DOS 4, 6 and 7; and under anaerobic conditions—DOS 3, 4 and 7 (Table 6).

Similar results were obtained in another experiment using the method of disk-diffusion bioassay. In this case, aliquots of FS from experimental and control treatments were applied to disks located on Petri dishes pre-seeded with test cultures (Figure 3). In the control treatment, after applying an aliquot of FS with added “Latrina” to the disks, zones of inhibition of the growth of test cultures (6–8 mm) were observed. In experimental treatments (FS aliquots were applied after incubation under aerobic conditions for 5 days in the presence of biological products), growth inhibition zones were either absent (in the case of DOS 4 and 7), or were insignificant (DOS 1–3, 5–6). This suggests that biological products almost completely eliminated the bactericidal effect of “Latrina”.

## 4. Discussion

There are currently a large number of bacterial products on the market for use as additives in portable toilets and septic tanks [58]. They are able to accelerate the biodegradation of human waste and provide disinfection. As a result of the microbiological activity induced by such biological products, the fecal masses are liquefied and reduced in volume, and their residues are pumped out more easily and with less frequency. Manufacturers of these biological products highlight, among the main advantages, their safety for the environment in contact with soil, water or air. However, the main drawback is also indicated: when using such bacterial products, it is necessary to avoid the simultaneous use of detergents, chlorine or antibacterial additives. These substances adversely affect the vital activity of microorganisms of bacterial products or even kill them. In our work, we propose a new approach: the creation of bacterial preparations based on microorganisms that are resistant to a certain group of contaminants and isolated from the environment in which these contaminants are present. We have demonstrated the effectiveness of this approach by preparing bacterial products that are effective in biodegrading FS containing biocidal agents such as DDAC and PHMG. Seven bacterial strains isolated from FS containing the biocide “Latrina” showed increased resistance to these antimicrobial agents. One of them, *A. faecalis* DOS7, was characterized by hyper-resistance, not stopping its growth even in the presence of various QACs and guanidine and biguanide derivatives of more than 500 ppm. Note that the MICs for these biocides of most bacteria are 100 or more times lower [56], which was also shown in our work. The study of bacterial preparations based on isolated bacteria showed their effectiveness for biodegradation of FS in the presence of the biocide “Latrina”. Based on two methods of assessment: according to the BOD_5_ test, as well as the disk-diffusion bioassay, we showed that the use of new bacterial products makes it possible to almost completely eliminate the toxic effect of the biocide in FS. This means that after pretreatment with bacterial products, FS become less harmful to the activated sludge of WWTP and can be disposed of without undesirable consequences. It should be noted that the isolated bacteria most probably do not use these biocides as a carbon source, but developing in its presence, they shift the pH of the medium to alkaline (final pH was around 8.7–8.9, data not shown), contributing to the decomposition of QAC and guanidine derivative [42]. Furthermore, apparently, the decomposition of toxic substances could be caused by the activity of *A. faecalis*, which are known for their ability to degrade various biocides. In particular, *A. faecalis* WY-01 can simultaneously remove phenol and ammonium, which can be used to treat specific wastewater containing these pollutants [59]. The genome of *A. faecalis* subsp. phenolicus MB207 has been shown to contain a copious amount of bioremediation genes, i.e., metal tolerance and xenobiotic degrading genes, suggesting that this strain is a prospective eco-friendly bacterium with numerous benefits for environment-related research [60]. A clearer explanation of the mechanisms of high resistance of *A. faecalis* DOS7 to QACs, as well as to guanidine and biguanide derivatives, could not be found in the literature. It can be assumed that it is similar to the resistance mechanisms found in some strains of *Pseudomonas aeruginosa*. Changes in fatty acid composition and in adapted cells have been reported [61,62,63,64] and are therefore believed to be a main adaptive defense mechanism against QACs. However, Mechin et al. (1999) showed that the alteration of membrane fatty acids could not be the only explanation for the increase in resistance, because the changes were evident from the first subculture and remained steady through the adaptation to higher concentrations of QACs [64]. Therefore, additional resistance mechanisms, such as efflux, degradation of disinfectant, slime formation or modified targets probably also contribute to the increased resistance [65].

It is also important that tests at the Research Center for Toxicology and Hygienic Regulation of Biopreparations confirmed that the isolated strains are not pathogenic for warm-blooded animals, and therefore can be used in bacterial products. Interestingly, *A. faecalis* DOS7, which showed the highest resistance to biocides, is a considered plant growth-promoting rhizobacteria (PGPB) because of its ability to produce indole acetic acid (IAA) and 1-aminocyclopropane-1-carboxylic acid (ACC) deaminase, solubilize phosphates, and fix atmospheric nitrogen [66,67]. *A. faecalis* is able to synthesize phytohormones that change the morphology and metabolism of plants, which leads to an improvement in the absorption of water and minerals [66]. In a number of works, it has been proposed to use *Alcaligenes* sp. as a biofertilizer to increase the growth and productivity of plants subjected to various types of stress, including salt stress [68]. It has been shown that inoculation with *A. faecalis* promotes the activation of plant growth parameters and yield by reducing the effect of salt stress on them [67,69]. Thus, bacterial products based on *A. faecalis* can be used not only to improve the biodegradation of FS and other organic waste contaminated with biocides, but also potentially in agriculture to increase plant productivity [70]. We believe that these bacterial products can also be used for the bioremediation of soils contaminated with QACs and guanidine and biguanide derivatives. The feasibility of new areas of use should be the focus of future research.

In the future, the following aspects are recommended for more in-depth study. First, it is necessary to study in detail the mechanisms of resistance of *A. faecalis* DOS7 to QACs and guanidine derivatives, which is extremely important for understanding the phenomenon of bacterial resistance and may be of applied importance in medicine. Secondly, for practical application, it is necessary to develop effective ways to use the obtained biological products for treatment of different types of waste. It would be interesting to investigate the effectiveness of co-application with other bacterial preparations (e.g., Novozymes, Roebic Laboratories, etc.) to obtain a synergistic effect. Third, using the approach proposed in this work, it is necessary to continue isolating microorganisms resistant to biocidal compounds, especially those that can use these compounds as a carbon source.

## 5. Conclusions

The treatment of municipal and industrial wastewater containing pollutants and toxic substances is relevant throughout the world. This work was devoted to the development of a new ecological approach to the disposal of fecal sludge containing biocidal agents. This approach is based on the use of biological products made from biocide-resistant bacterial strains. Thus, seven strains of bacteria with high resistance to widely used biocidal agents, DDAC and PHMG, were isolated from fecal sludge. The values of the minimum inhibitory and bactericidal concentrations of biocides for the isolated strains were 4.5–10 times higher than for the collection microorganisms. Biological products prepared using these strains demonstrated a positive effect on the processes of biodegradation of fecal sludge contaminated with DDAC and PHMG. In addition, results of BOD_5_ tests and disk-diffusion bioassay confirmed that isolated strains contribute to reducing the toxicity of biocidal agents in fecal sludge. The bacterium *A. faecalis* DOS7 was found to be particularly resistant to “Latrina” (a chemical toilet additive containing DDAC and PHMG), the minimum inhibitory concentration of which was almost 30 times higher than recommended for ESTC. Due to hyper-resistance, non-pathogenicity, and a potential beneficial trait such as plant growth-promoting ability, *A. faecalis* DOS7 can be a very promising strain for use in various biological products for the treatment of wastewater and bioremediation of soils contaminated with biocides, as well as in agriculture to increase plant productivity.

Further research will be aimed at selecting the most technologically advanced and cost-effective conditions for the use of biological products for the detoxification of contaminated FS before its disposal at treatment facilities. It will also be interesting to study the ability of the biological product based on *A. faecalis* DOS7 to stimulate the growth and productivity of agricultural plants. The approach tested in this work will be further used to isolate microorganisms resistant to other types of biocides.

## 6. Patents

Patent RU 2 751 795 C1 resulted from the work reported in this manuscript.

## Figures and Tables

**Figure 1 biology-11-01332-f001:**
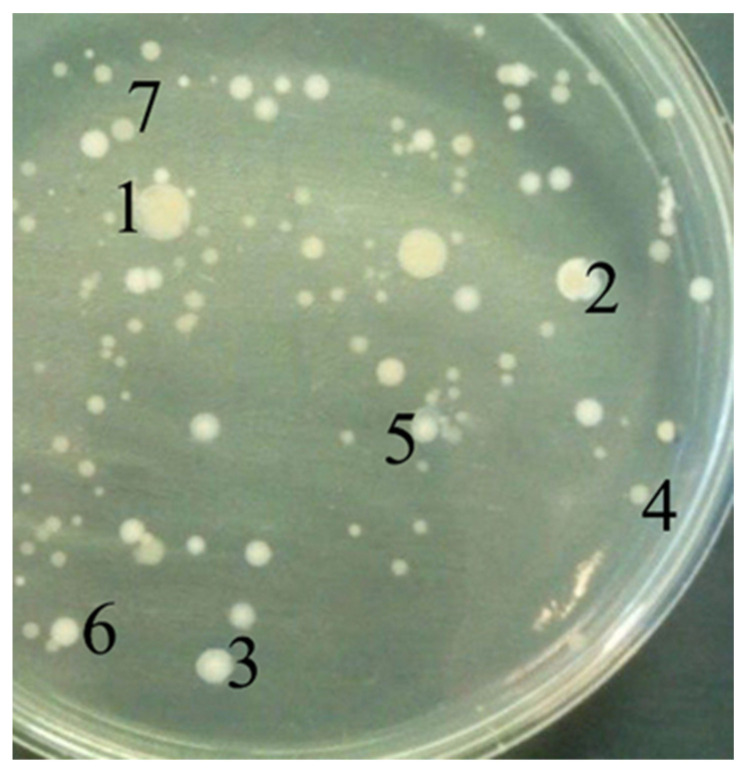
View of colonies selected for further work.

**Figure 2 biology-11-01332-f002:**
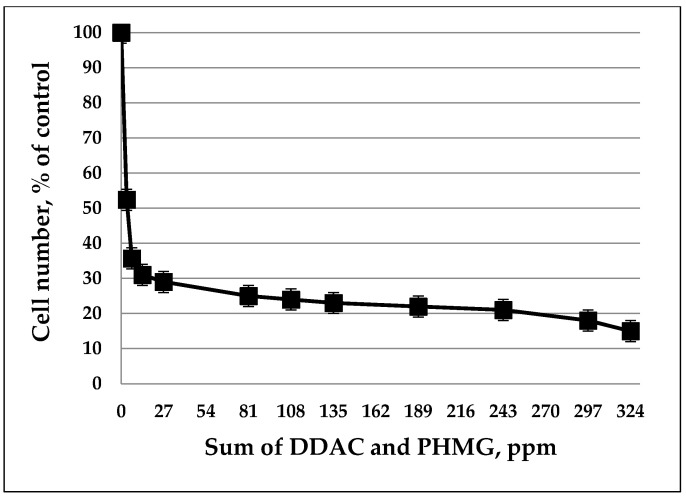
Effect of DDAC and PHMG on the growth of *A. faecalis* DOS7.

**Figure 3 biology-11-01332-f003:**
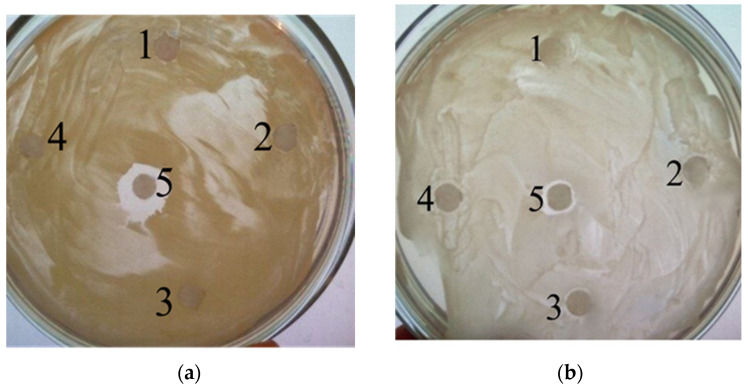
Disk-diffusion bioassay using *M. luteus* NCIMB 13267 (**a**) and *Y. lipolytica* 367-2 (**b**) as test cultures: 1—FS with DOS 1; 2—FS with DOS 6; 3—FS with DOS 4; 4—FS with DOS 7; 5—Control (FS without the addition of a biological product). The figure shows typical bioassay photos. DOS 2, 3 and 5 showed similar results (data not shown).

**Table 1 biology-11-01332-t001:** Genbank accession numbers of isolated strains.

No	Isolated Strains	Genbank Accession Number
1	*Paenalcaligenes* sp. DOS1	OL454899
2	*Staphylococcus pasteuri* DOS2	OL454901
3	*Enterococcus faecium* DOS3	OL454902
4	*Micrococcus endophyticus* DOS4	OL454903
5	*Enterococcus casseliflavus* DOS5	OL454904
6	*Bacillus subtilis* DOS6	OL454905
7	*Alcaligenes faecalis* DOS7	OL454906

**Table 2 biology-11-01332-t002:** MIC and MBC of sum of DDAC and PHMG (as part of “Latrina”) for isolated and some collection strains.

No	Microorganism	MIC, ppm	MBC, ppm
	**Isolated Strains**		
1	*Paenalcaligenes* sp. DOS1	150	400
2	*Staphylococcus pasteuri* DOS2	16	32
3	*Enterococcus faecium* DOS3	10	15
4	*Micrococcus endophyticus* DOS4	45	70
5	*Enterococcus casseliflavus* DOS5	10	15
6	*Bacillus subtilis* DOS6	35	70
7	*Alcaligenes faecalis* DOS7	1350	2700
	**Collection Strains**		
8	*Staphylococcus aureus* 209P	3.5	7.0
9	*Pseudomonas aeruginosa* 4.8.1	105	140
10	*Bacillus subtilis* 720	3.5	7.0
11	*Enterococcus faecium* M	2.0	4.0
12	*Micrococcus luteus* NCIMB 13267	7.0	14.0

**Table 3 biology-11-01332-t003:** The number of viable cells grown in the presence of biocide “Latrina” (% of the cell number in the control population without biocide).

Isolated Strains	The Total Concentration of DDAC and PHMG in the Nutrient Medium, ppm			
	0	3.5	7	14
*Paenalcaligenes* sp. DOS1	(2.2 ± 0.4)·10^9^ (100)	(2.7 ± 0.4)·10^8^ (12.3)	(1.9 ± 0.4)·10^8^ (8.6)	(1.1 ± 0.4)·10^8^ (5.0)
*Staphylococcus pasteuri* DOS2	(1.5 ± 0.1)·10^9^ (100)	(3.6 ± 0.2)·10^8^ (24.0)	(1.9 ± 0.1)·10^8^ (12.6)	(6.2 ± 0.3)·10^7^ (4.1)
*Enterococcus faecium* DOS3	(1.0 ± 0.2)·10^9^ (100)	(1.5 ± 0.1)·10^8^ (15.0)	(5.9 ± 0.1)·10^7^ (5.9)	(3.4 ± 0.1)·10^7^ (3.4)
*Micrococcus endophyticus* DOS4	(3.2 ± 0.3)·10^9^ (100)	(6.7 ± 0.4)·10^8^ (20.9)	(1.7 ± 0.1)·10^8^ (5.3)	(7.7 ± 0.4)·10^7^ (2.4)
*Enterococcus casseliflavus* DOS5	(1.2 ± 0.1)·10^9^ (100)	(1.1 ± 0.1)·10^8^ (9.2)	(4.7 ± 0.3)·10^7^ (3.9)	(3.7 ± 0.2)·10^7^ (3.1)
*Bacillus subtilis* DOS6	(1.9 ± 0.1)·10^9^ (100)	(5.2 ± 0.2)·10^8^ (27.4)	(1.1 ± 0.1)·10^8^ (5.7)	(6.2 ± 0.4)·10^7^ (3.2)
*Alcaligenes faecalis* DOS7	(4.2 ± 0.4)·10^9^ (100)	(2.2 ± 0.1)·10^9^ (52.4)	(1.5 ± 0.1)·10^9^ (35.7)	(1.3 ± 0.1)·10^9^ (31.0)

**Table 4 biology-11-01332-t004:** MIC and MBC of some individual GACs and guanidine and biguanide derivatives for *A. faecalis* DOS7 in comparison to biocide “Latrina”.

No	Biocide	MIC, ppm	MBC, ppm
1	DDAC	900	900
2	ADBAC	600	600
3	PHMG	500	500
4	CHG	1000	2000
5	“Latrina” (DDAC + PHMG)	1350	2700

**Table 5 biology-11-01332-t005:** Biotechnological indicators of FS biodegradation processes under aerobic and anaerobic conditions.

Biological Product	Aerobic Conditions		Anaerobic Conditions	
	O_2_ Consumption Rate, mmol O_2_/mL FS/Day (in % of Control)	The Number of Viable Cells, CFU/mL	CO_2_ Formation Rate mmol CO_2_/mL FS/Day (in % of Control)	The Number of Viable Cells, CFU/mL
Control	0.0092 (100)	(1.4 ± 0.1)·10^6^ (100)	0.0004 (100)	(1.2 ± 0.1)·10^6^ (100)
DOS1	0.0190 (206.5)	(6.5 ± 0.2)·10^7^ (4643)	0.0082 (2050)	(1.1 ± 0.1)·10^7^ (916.7)
DOS2	0.0190 (206.5)	(1.4 ± 0.1)·10^7^ (1000)	0.0069 (1725)	(3.4 ± 0.1)·10^6^ (283.3)
DOS3	0.0197 (214.1)	(2.2 ± 0.2)·10^7^ (1571)	0.0086 (2150)	(4.2 ± 0.2)·10^6^ (350.0)
DOS4	0.0193 (209.8)	(5.1 ± 0.1)·10^7^ (3643)	0.0089 (2225)	(6.1 ± 0.1)·10^7^ (5083.3)
DOS5	0.0194 (210.8)	(6.2 ± 0.3)·10^7^ (4429)	0.0082 (2050)	(4.1 ± 0.3)·10^7^ (3416.7)
DOS6	0.0198 (215.2)	(6.4 ± 0.2)·10^7^ (4571)	0.0077 (1925)	(5.0 ± 0.2)·10^7^ (4167.7)
DOS7	0.0205 (222.8)	(5.9 ± 0.2)·10^8^ (42,143)	0.0079 (1975)	(8.5 ± 0.2)·10^7^ (7083.3)

**Table 6 biology-11-01332-t006:** BOD_5_ (in mg O_2_/L) of FS in control and experimental treatments incubated for 5 days under aerobic or anaerobic conditions.

Treatment	Aerobic Process	Anaerobic Process
Control 1: FS without added “Latrina“	5690	5690
Control 2: FS with added “Latrina”	2027	4500
FS with added “Latrina” + DOS1	4310	4880
FS with added “Latrina” + DOS2	4650	4930
FS with added “Latrina” + DOS3	4730	5010
FS with added “Latrina” + DOS4	5120	5140
FS with added “Latrina” + DOS5	4950	4850
FS with added “Latrina” + DOS6	5600	4900
FS with added “Latrina” + DOS7	5630	5250

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
