# Peer review of "Didecyldimethylammonium Chloride- and Polyhexamethylene Guanidine-Resistant Bacteria Isolated from Fecal Sludge and Their Potential Use in Biological Products for the Detoxification of Biocide-Contaminated Wastewater Prior to Conventional Biological Treatment"

_biology, 2022, doi:10.3390/biology11091332_

Round 1
Reviewer 1 Report
In the present paper, the authors have investigated the capability of biological products from biocide-resistant bacterial strains isolated from faecal sludge to reduce the toxicity of faecal sludge itself, which contains biocides based on quaternary ammonium compounds and guanidine derivatives, prior disposal to biological treatment plants. Among isolated strains, Alcaligenes faecalis DOS7 was found to be particularly resistant to the biocides and appears the most promising in faecal sludge treatment.
Overall, the manuscript is well-written and easy to read. The topic is of high interest and has immediate application for the detoxification of faecal sludge before itsimmission in biological conventional treatment.
I have no major recommendations for the authors. I think the paper just requires proof-reading for English, because just minor mistakes are present. I only want to bring to authors' attention lines 449-451 where there is a prhase in Russian in the middle of the paragraph, and therefore the meaning was not clear.
Kind regards
Author Response
Dear Reviewer,
We thank you for your positive response and careful consideration of our manuscript. We translated the Russian phrase, and now it is clearer. The details are as follows:
“Also, apparently, the decomposition of toxic substances could be caused by the activity of A. faecalis, which are known for their ability to degrade various biocides.”
Reviewer 2 Report
When isolation of toxic resistance microorganisms from anthropogenic active site, nutrient or minimum agar plate including referenced compounds at being threshold value would be effective to achieve the aim.
L332 This may be due to the presence…….
Is not this logic unclear. DDAC and PHMG both categorized in surfactant?
Author Response
- When isolation of toxic resistance microorganisms from anthropogenic active site, nutrient or minimum agar plate including referenced compounds at being threshold value would be effective to achieve the aim.
Dear Reviewer, we thank you for this and another good question below. In fact, we tried to go the way you mentioned and plated FS samples on agar media containing some amount of DDAC and PHMG. However, at even a relatively low concentration of biocides in the agar medium, the growth of microorganisms was inhibited. This may be due to the multiple stress (stress of the new environment, oxidative stress, and action of biocides) to which the microorganisms were subjected. By decreasing the concentration of biocides to very low levels, we did not observe a difference with the control dishes. Based on the results of this experiment, we decided to isolate the most rapidly growing and widespread colonies on control plates without biocides, and then evaluate their resistance to DDAC and PHMG.
- L332 This may be due to the presence…….
Is not this logic unclear. DDAC and PHMG both categorized in surfactant?
It was mentioned in Section 2.1, that “The composition of "Latrina" included: DDAC (0.24%) and PHMG (6.5%) (total 6.74%), surfactants, perfume and water”
If we understand the reviewer's question correctly, then PHMG and especially DDAC also have the ability to lower surface tension. At the same time, in the context of the work, these components are considered as disinfectants, and, accordingly, their main function in "Latrina" is a biocidal agent, and not surfactants (which were not revealed by the manufacturer).
However, for clarity, we have added additional explanations in Section 3.2. The details are as follows:
“This may be due to the presence of other surfactants in the composition of "Latrina" (in addition to PHMG and especially DDAC, which also have the ability to lower surface tension), since it has been shown that in the presence of anionic surfactants, QACs become less effective [57]”
Thank you again for your positive and constructive comments and suggestions on our manuscript. We hope you will find our revised manuscript acceptable for publication.
Reviewer 3 Report
Section 2.2: What is the basis for selecting colonies?
What measures were taken to avoid contamination?
BOD5 how much volume of liquid was taken? Did it disrupt the experimental process?
Figure 2, what about deviations?
The conclusion section should be expanded, and the future scope of the work is to be discussed.
Author Response
- Section 2.2: What is the basis for selecting colonies?
In this work, we selected 7 colonies, from which, as it seemed to us, it was promising to isolate pure cultures of microorganisms resistant to biocides. These colonies were the first to appear and were numerous.
In section 3.1, we specified the principle of colony selection, the details are as follows: “For further work, 7 different colonies were selected, which were the most common and the first to appear on the agar medium (Fig. 1)”
2.What measures were taken to avoid contamination?
All microbiological experiments were performed under sterile conditions in a laminar flow box. This details were included in the materials and methods section 2.2:
“All microbiological experiments were performed under sterile conditions in a laminar flow box”
- BOD5 how much volume of liquid was taken? Did it disrupt the experimental process?
To determine BOD5, 35 ml of the sample was taken. This did not disturb the experimental process, since additional replicates were made for each determination of BOD5. We have included this information in Section 2.8. The details are as follows:
“After 5 days of incubation, FS samples (35 mL) were taken to determine the five-day biochemical oxygen demand (BOD5). This did not disturb the experimental process, since additional replicates were made for each determination of BOD5.”
- Figure 2, what about deviations?
Deviations are indicated on the graph. However, they are small, so they are not noticeable for every measured point.
- The conclusion section should be expanded, and the future scope of the work is to be discussed.
Thank you very much for your suggestion. We've added additional discussion to the conclusion section.
Thank you again for your positive and constructive comments and suggestions on our manuscript. We hope you will find our revised manuscript acceptable for publication.